applied mathematics/statistics/ecology

critical slowing down, early warning signals, dynamical systems

**Author for correspondence:**
Amin Ghadami
e-mail: aghadami@umich.edu

# Practical guide to using Kendall's $\tau$ in the context of forecasting critical transitions

Shiyang Chen, Amin Ghadami and Bogdan I. Epureanu

Department of Mechanical Engineering, University of Michigan, Ann Arbor, MI, USA

SC, 0000-0001-8974-3705; AG, 0000-0001-5883-4153

Recent studies demonstrate that trends in indicators extracted from measured time series can indicate an approach of an impending transition. Kendall's $\tau$ coefficient is often used to study the trend of statistics related to the critical slowing down phenomenon and other methods to forecast critical transitions. Because statistics are estimated from time series, the values of Kendall's $\tau$ are affected by parameters such as window size, sample rate and length of the time series, resulting in challenges and uncertainties in interpreting results. In this study, we examine the effects of different parameters on the distribution of the trend obtained from Kendall's $\tau$, and provide insights into how to choose these parameters. We also suggest the use of the non-parametric Mann–Kendall test to evaluate the significance of a Kendall's $\tau$ value. The non-parametric test is computationally much faster compared with the traditional parametric auto-regressive, moving-average model test.

## 1. Introduction

Complex systems might undergo abrupt transitions from one stable state to another [1–7]. Such a transition can be undesirable, leading to various types of stability issues and possible catastrophic consequences. Specifically, regime shifts in ecological systems have received growing attention as the cumulative human impact on the environment has increased the risk of ecological regime shifts [4]. The prediction of critical transitions faces significant challenges because changes in the equilibrium state of the system are generally small prior to transitions. Recent studies demonstrate that several indicators based on changes in ecological time series can indicate that the system is approaching an impending transition [8–12]. These indicators, called early warning indicators, are statistical measures that reveal proximity to a tipping point developed based on the slowing down phenomenon [13]. When a dynamical system approaches a tipping point, perturbations lead to a long transient before the system reaches its stable state, a

phenomenon known as slowing down. As a consequence of the slowing down phenomenon, an increase in the values of certain statistics, i.e. variance and lag-1 autocorrelation of stochastic fluctuations of the system, have been observed prior to critical transitions in numerous theoretical and experimental complex systems [2,3].

To probe for indications of critical slowing down prior to a transition, the trend of the extracted warning indicators is monitored as system parameters gradually change [14]. An increasing (positive) trend in the values of early warning indicators over time is considered as a sign of approaching a transition. Such a trend, however, needs to be quantified to allow one to analyse the changes in the system dynamics. In addition, identifying the trend of early warning signals might not be trivial due to stochastic fluctuations in the reported values of early warning signals over time. As a result, Kendall's $\tau$ coefficient is often used to quantify the trend of statistics related to the critical slowing down phenomenon [15,16]. Kendall's $\tau$ is a measure of the correlation between the rank order of the observed values and their order in time [16]. A positive Kendall's $\tau$ typically means a monotonic increase in the data. This metric has been used to quantify the trend of early warning signals and risk of impending transitions in a variety of systems, particularly in computational [7,8,14] and empirical [17–19] ecological and climate systems. However, interpreting the values of Kendall's $\tau$ is challenging and requires careful considerations. First, there exists a probability distribution over the possible values of Kendall's $\tau$ corresponding to each measurement. Hence, a positive Kendall's $\tau$ does not guarantee that the system is moving towards a transition unless its significance is confirmed [20,21]. Even for a signal measured from a stationary system, one might obtain a positive value for Kendall's $\tau$. In addition, the values of Kendall's $\tau$ are affected by parameters such as window size, sample rate and the length of the time series corresponding to data collection and analysis steps [14,20,21]. Hence, one needs to examine the effects of different parameters on the distribution of the trend statistic Kendall's $\tau$. Without such a study and detailed understanding of the statistical significance of the estimated Kendall's $\tau$ from measured time series, it would be difficult to conclude if a detected warning signal is a false alarm or not.

A number of parametric and non-parametric methods have been proposed to understand the significance of Kendall's $\tau$ values obtained from time series [8,21]. Parametric tests can be more powerful, but require more information about the system. By contrast, non-parametric trend tests, such as the Mann–Kendall test, require only that data be independent and tolerate outliers [22]. However, given that early warning signals for critical transitions are estimated using sliding windows across the time series, serial correlations are unavoidable and pose a challenge on the application of non-parametric tests in this context.

This study aims to identify the factors that affect estimated Kendall's $\tau$ statistics, and provide guidance for the interpretation of estimated Kendall's $\tau$ values. Highlighting the need to perform a significance test on the estimated Kendall's $\tau$ values, we compare the use of parametric and non-parametric tests in identifying the significance of Kendall's $\tau$ obtained from surrogate ecological time-series measurements and discuss the benefits and drawbacks of each method in the context of forecasting critical transitions. In particular, we discuss the advantages of a modified Mann–Kendall test as a non-parametric test that accounts for serial correlations in the approximated early warning signals of critical transitions. Results of this study may improve the reliability of predictions made about the risk of critical transitions in complex systems based on early warning signals.

## 2. Effects of data availability and data processing on Kendall's $\tau$ statistics

In this section, we use a simple example system to highlight how different parameters of the analysis affect the distribution of Kendall's $\tau$ for systems either facing or not facing a critical transition. Simulation data are obtained from the harvesting model [23]

$$\mathrm{d}x = \left(rx\left(1 - \frac{x}{K}\right) - c\frac{x^2}{x^2 + 1}\right)\mathrm{d}t + \sigma\,\mathrm{d}W, \tag{2.1}$$

where $x$ is the amount of biomass, $K$ is the carrying capacity, $r$ is the maximum growth rate, $c$ is the maximum grazing rate and $\sigma$ is the standard deviation of the white noise $\mathrm{d}W$. Values of the parameters (except for the bifurcation parameter $c$) are selected as $K = 10$, $r = 1$ and $\sigma = 0.01$. In the case that $\sigma = 0$, equation (2.1) is deterministic and has one or more equilibria depending on the parameter $c$ which satisfy $\mathrm{d}x/\mathrm{d}t = 0$. The bifurcation diagram for this system is depicted in figure 1. As harvesting rate increases, biomass gradually decreases up to a critical threshold

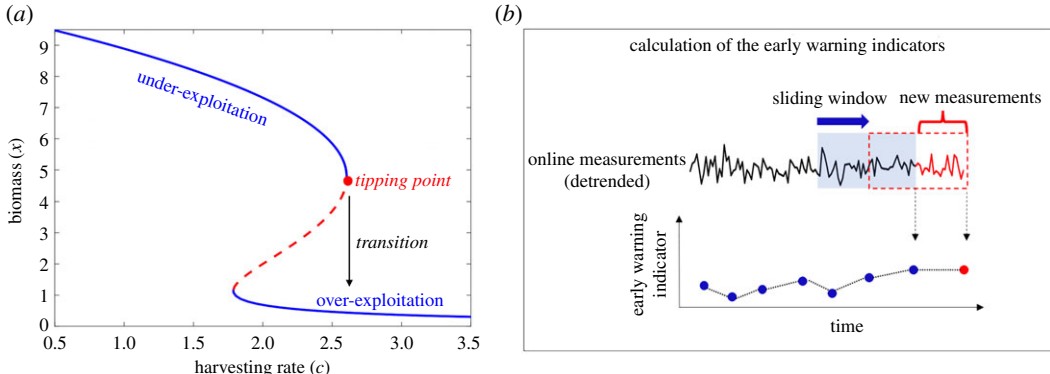

**Figure 1.** (*a*) Bifurcation diagram of the spatial harvesting model. As harvesting rate *c* increases, biomass gradually decreases up to a critical threshold where the under-exploited biomass undergoes a critical transition to the alternative over-exploited state. (*b*) Schematic of calculation of early warning signals. Kendall's $\tau$ is used to identify the trend of early warning signals over time. A significant positive trend alarms approaching to a transition.

($c_{\text{critical}} = 2.60$) where the biomass undergoes a critical transition. When the harvesting rate passes this critical threshold, the resource collapses from under-exploited to the alternative over-exploited state.

When the system approaches its tipping point, one of the eigenvalues approaches zero and becomes less negative and thus the return to the equilibrium becomes slower, a phenomenon known as critical slowing down. In systems with stochastic effects in their dynamics, early warning indicators have been proposed [2,14,24] that estimate this return rate indirectly, such as the variance and autocorrelation of fluctuations in a time series of the observations of the state of a system. An increasing (positive) trend in the values of early warning indicators over time is considered as a sign of approaching a transition. One of the widely used tests for detecting trends in the approximated early warning signals is the non-parametric trend statistic Kendall's $\tau$, which is a measure of the correlation between the rank order of the observed values and their order in time [16].

Assume that $(x_i, y_i)$, $i = 1, \dots, n$ are a set of $n$ observations of random variables $X$ and $Y$, where $X$ and $Y$ may be ranks or variables. Kendall's rank correlation measure is defined as follows, representing the probability of concordance minus the probability of discordance for a pair of observations $(x_i, y_i)$ and $(x_j, y_j)$ chosen randomly from the sample [15,16]

$$\tau = \frac{2}{n(n-1)} \sum_{i<j} \text{sign}(x_j - x_i) \, \text{sign}(y_j - y_i). \tag{2.2}$$

If the values in $Y$ are replaced with the time order of the time series $X$, i.e. 1, 2, …, $n$, the test can be used as a trend test, which is a common approach in quantifying the trend of early warning indicators of critical transitions.

To use Kendall's $\tau$ to detect critical slowing down, target statistics such as the variance or the autocorrelation are first calculated using a moving window. This sequence of statistics is then used to calculate Kendall's $\tau$, which is used to evaluate the trend of the statistics and make a decision about the system. However, interpreting the values of Kendall's $\tau$ requires careful consideration, because Kendall's $\tau$ values are affected by several factors and there exists a probability distribution over the possible values of Kendall's $\tau$ corresponding to each measurement.

For a sequence of independent and randomly ordered data, i.e. when there is no trend or serial correlation structure among the observations, the trend statistic Kendall's $\tau$ should tend to a normal distribution for a large number $n$ of observations in the sequence. The normal distribution has mean zero and variance given by Kendall [16]

$$\text{var}(\tau) = \frac{2(2n+5)}{9n(n-1)}. \tag{2.3}$$

The approximated Kendall's $\tau$ can be evaluated using this normal distribution as a null hypothesis to identify the significance of the measured Kendall's $\tau$. Generic early warning signals, however, are calculated using a sequence of statistics that are obtained from the time series using a moving window. Positive correlation among the observations increases the chance of obtaining a large

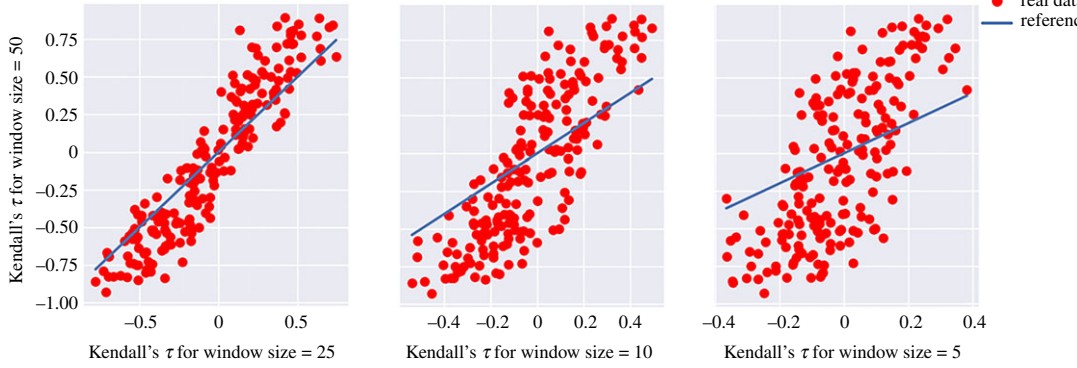

**Figure 2.** Comparison between Kendall's $\tau$ calculated using a larger window size and a smaller window size. Each window size is represented as a percentage of the total length of the time series.

Kendall's $\tau$, even in the absence of an actual trend [22]. In particular, the choice of parameters, such as window size and sample rate, affects the correlation in data and consequently the distribution of Kendall's $\tau$ that causes the distribution approximated by equation (2.3) to deviate from the real distribution.

Below, we first study the effect of the window size and the number of observations on the distribution of Kendall's $\tau$ approximated for the harvesting model of equation (2.1). We show that these parameters significantly affect the distribution of trends of the early warning signals. Next, we discuss parametric and non-parametric approaches to efficiently analyse the significance of approximated trends of the warning signals.

## 2.1. Window size

The choice of window size has a large influence on the distribution of Kendall's $\tau$. This is because a positive serial correlation exists when two consecutive moving windows have an overlap. This correlation is even stronger as the size of the moving window increases.

To show this, we collected 400 time series from the harvesting model in equation (2.1) with a fixed parameter value $c = 1.1$ and calculated Kendall's $\tau$ using a different window size for each time series. All time series have the same total sample size of 12 000. To calculate Kendall's $\tau$, similar to the procedure depicted in figure 1, a window size is selected and is moved with a small stride of 100 over the measured data. The variance of data in the sliding window is calculated over time as the early warning signal. Finally, the trend of the measured warning signal is calculated using Kendall's $\tau$ as presented in equation (2.2). In this section, we study the effect of the size of the window on Kendall's $\tau$ values that are obtained. To facilitate comparison, the window sizes are chosen as a percentage of the length of the time series, namely 50%, 25%, 10% and 5%. Figure 2 shows the relationship between Kendall's $\tau$ calculated using a smaller window size and a large window size. Each dot in the plot represents a result obtained from one time series calculated by solving equation (2.1). Results show that the curve takes an S shape as the difference between window sizes increases. This means that a large window size will inflate the value of Kendall's $\tau$ calculated from the same time series/data.

The inflation of Kendall's $\tau$ for large window sizes can also be observed using the distribution plot shown in figure 3. As the window size increases, the distribution of Kendall's $\tau$ becomes flatter and farther away from the normal distribution, with variance given by equation (2.3).

Therefore, the same Kendall's $\tau$ value has a completely different meaning with a different window size. A 90% percentile Kendall's $\tau$ value when the window size is 5% of the length of the time series is only 60% percentile Kendall's $\tau$ value when the window size is 50% of the length. In addition, one should note that for data recorded under the same parameter condition, Kendall's $\tau$ can take a wide range of values, both positive and negative, which might result in a false positive or a false negative alarm. Thus, merely calculating Kendall's $\tau$ is not enough to decide the probability of a critical transition, and a hypothesis test is necessary. A hypothesis test based on a modified Mann–Kendall approach is introduced in §3.

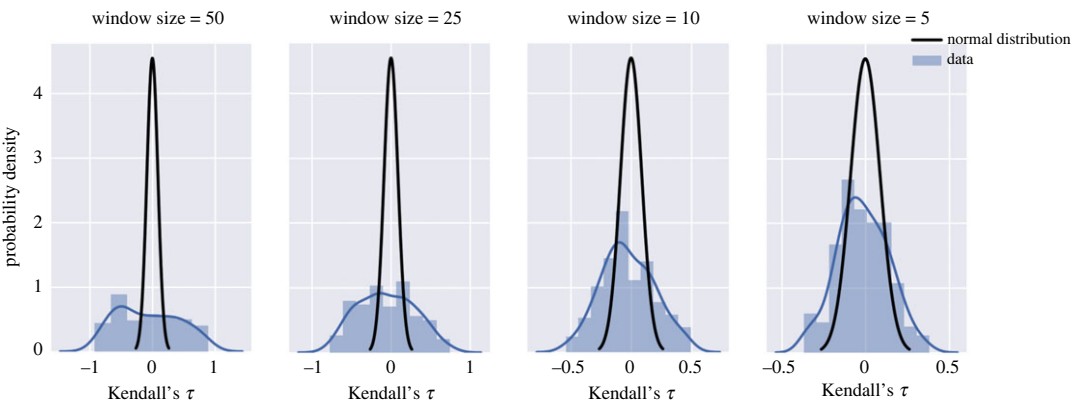

**Figure 3.** Distribution of Kendall's $\tau$ calculated using different window sizes. Each window size is represented as a percentage of the total length of the time series.



**Figure 4.** Distribution of Kendall's $\tau$ when 20 000, 2000, 200 and 100 observations are used.

## 2.2. Number of observations

The number of observations in each of the datasets is also important. That is because all statistics will have a larger estimation error when only a limited number of observations are available. Moreover, it is harder to detrend the time-series data when only a limited number of observations are available. Improper detrending may remove the important low-frequency information, leaving behind only high-frequency random noise.

To understand how the sampling rate can affect results, one-time-series data is obtained from the harvesting model in equation (2.1). The time-series data is then down-sampled to obtain another time series with a smaller number of observations. The effect of the number of available data points can be observed in the distribution of Kendall's $\tau$ when different numbers of observations are available. We again collected 400 time series of equal time length from the harvesting model (equation (2.1)), with parameter $c$ continuously changing with time from $c = 0.6$ to $c = 1.1$, generating non-stationary time series approaching a critical transition. Each time series has 20 000 observations. These 400 time series are then down-sampled to obtain time series with 2000, 200 and 100 observations. The resulting time series are then detrended to study the statistics around the system equilibrium, a standard procedure in the studies of early warning indicators of critical transitions [8]. The distribution of Kendall's $\tau$ is shown in figure 4. When there are at least 2000 observations, the distribution of Kendall's $\tau$ is skewed toward the right, which is correct because the system is approaching the critical transition. However, when there are only 200 observations or fewer, the distribution becomes almost symmetric about zero, which is associated normally with random signals. Therefore, it is important to have enough observations in the data when Kendall's $\tau$ is used as an early warning signal, especially when the equilibrium of the system is changing as the system approaches the critical transition and detrending is necessary.

## 3. Parametric and non-parametric tests to evaluate the significance of Kendall's $\tau$ values

Because Kendall's $\tau$ is affected by several parameters, as discussed in the previous section, merely observing the values of Kendall's $\tau$ does not always reveal the desired information about the system.

One should evaluate the significance of the approximated Kendall's $\tau$ values to identify whether the system is at risk of critical transitions. A number of tests have been proposed to understand the significance of a certain value of Kendall's $\tau$ in the literature [8,14,25,26]. These approaches can be categorized as parametric and non-parametric tests. In these approaches, the goal is to identify the null hypothesis that there is no trend in the data, and identify the significance of the approximated Kendall's $\tau$ assuming the null hypothesis is true. In this section, we review the parametric and non-parametric tests and compare them in the context of early warning signals. Particularly, we introduce the modified Mann–Kendall trend test [22] as a potential and efficient method to evaluate the significance of Kendall's $\tau$ values of early warning signals where data availability is limited. Unlike common methods used in the context of early warning signals of critical transitions to evaluate the significance of approximated Kendall's $\tau$, the modified Mann–Kendall does not need a large number of randomized surrogate datasets and can approximate the distribution of Kendall's $\tau$ values using a single time series.

## 3.1. Non-parametric Mann–Kendall trend test

The non-parametric Mann–Kendall test is commonly employed to detect monotonic trends in time series. The null hypothesis for the traditional Mann–Kendall trend test (equation (2.3)), however, is that the data are independent, randomly ordered, without serial correlation structure among the observations. However, in many real situations, the observed data are autocorrelated, which can result in misinterpretation of trend test results. In particular, in the context of early warning signals, this null hypothesis is not rigorously obeyed for the series of statistics such as the standard deviation or the autocorrelation that are obtained from a time series using a moving window. As a result, using the standard Mann–Kendall test does not lead to reliable results in identifying the significance of the trend of the early warning signals [21].

Hamed & Rao [22] point out that a modified Mann–Kendall trend test can be used to study data with a serial correlation structure. In the modified test, the null hypothesis is that there is no trend in the data, but there can be autocorrelation, addressing the challenge existing in the studies of early warning signals. If the null hypothesis is true, Kendall's $\tau$ should follow a normal distribution with mean 0, and variance given by Hamed & Rao [22]

$$\text{var}(\tau) = \frac{2(2n+5)}{9n(n-1)}\left(1 + \frac{2}{n(n-1)(n-2)}\sum_{i=1}^{n-1}(n-i)(n-i-1)(n-i-2)\rho_S(i)\right), \quad (3.1)$$

where $\rho_S(i)$ is the autocorrelation of the ranks of the observation and $n$ is the number of observations.

To better understand this, we compared the distribution of Kendall's $\tau$ with the normal distribution with variance calculated using equation (2.3) and the modified distribution with variance calculated using equation (3.1). In this example, we use 200 distinct time series generated using the harvesting model at parameter $c = 1.1$ (equation (2.1)). For each time series, we calculated the variance as the early warning signal using a moving window. The window size and number of observations are varied and the values of Kendall's $\tau$ are then calculated for each set of obtained warning signals. The distribution of Kendall's $\tau$ obtained using the generated time series is shown in figure 5 revealing that the real distribution is much flatter than the normal distribution due to the positive correlation in data. Next, a single time series is used to calculate the varience of the normal distribution obtained by the modified Mann–Kendall test (i.e. equation (3.1)). Results shown in figure 5 show that the modified distribution is much closer to the real distribution. In practice, one may typically only have one time series. Therefore, the real distribution is not available. In this case, we can use the available time series to calculate the modified distribution of Kendall's $\tau$, and use the distribution to calculate the percentile of the obtained Kendall's $\tau$ value.

In the context of early warning signals of critical transitions, the serial correlations in data result mainly from overlapping windows that are used in the procedure of approximating warning signals. Here, we have studied the effects of using different window overlaps on the approximated distribution of Kendall's $\tau$ using Mann–Kendall's test and the modified test. Results of this analysis are demonstrated in figure 6. We studied four different scenarios where the total number of samples is 1200, the window size is kept at 10% (120 samples), and strides are 10, 20, 50 and 100. Results show that when there are large overlaps (10 or 20 sample stride), the normal Mann–Kendall test underestimates the variance in Kendall's $\tau$, while the modified Mann–Kendall test still accurately captures the distribution. For little to no overlaps between the windows (50 or 100 sample stride),

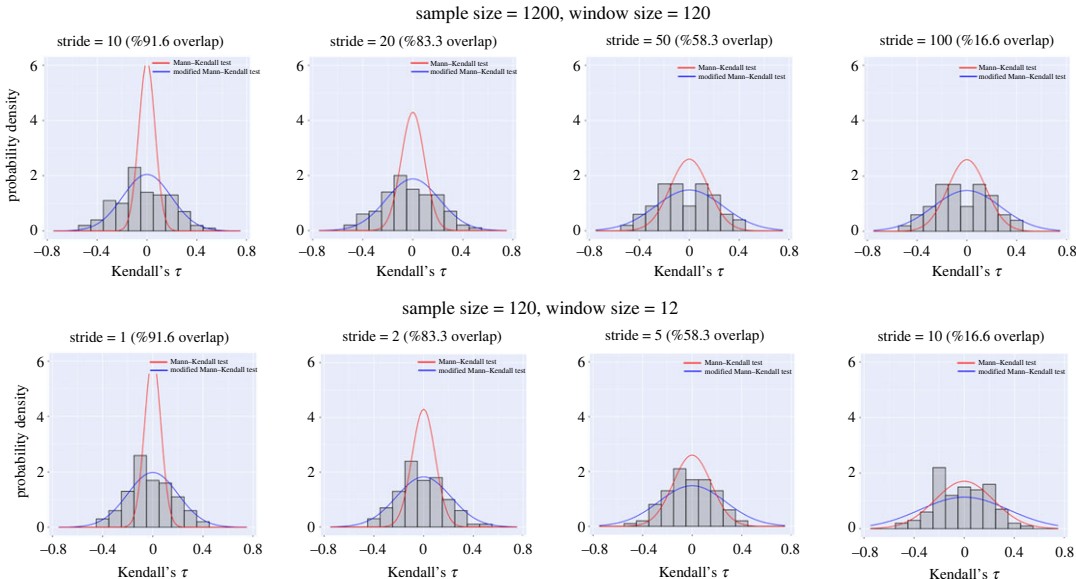

**Figure 5.** Comparison between the distribution of Kendall's $\tau$ approximated using data, the Mann–Kendall test and the modified Mann–Kendall test for different window sizes and number of observations. The control parameter is fixed at $c = 1.1$.

**Figure 6.** Comparison between the distribution of Kendall's $\tau$ approximated using data, the Mann–Kendall test and the modified Mann–Kendall test for different window overlaps and number of observations. The control parameter is fixed at $c = 1.1$.

both regular and modified Mann–Kendall tests show good approximation to the true distribution. We further studied the situation when fewer samples (i.e. 120 samples) are available, and similar results were observed. While decreasing the overlap between subsequent windows improves the accuracy of

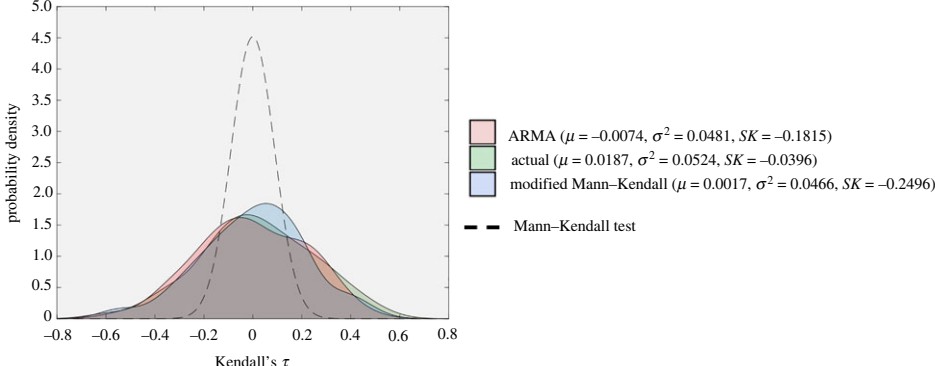

**Figure 7.** Distribution of Kendall's $\tau$ calculated using the harvesting model, the fitted ARMA model and the modified Mann–Kendall test. The mean ($\mu$), variance ($\sigma^2$) and skewness (*SK*) for each distribution is demonstrated.

the Mann–Kendall test, one should note that in the practical applications of early warning signals, choosing windows with large overlaps are inevitable. The reason for this is that new data should be analysed as soon as possible to detect potential transitions early enough. Taking a long stride puts the system at risk because changes in the system dynamics might not be noticed during this period. In addition, using non-overlapping windows requires a lot of data which might not be available in many practical scenarios.

Results of figures 5 and 6 highlight the importance of performing a significance test using a modified version of the null hypothesis, such as a modified Mann–Kendall test, in order to capture the true distribution of Kendall's $\tau$ with a higher accuracy particularly when using a large window size or a large overlap.

## 3.2. Parametric tests

Parametric tests [8,21] are a group of tests that have been proposed to study the significance of Kendall's $\tau$ values. These methods use a general model to fit the data, and generate artificial data using the model to understand how significant the trend statistic value is. Dakos *et al.* [8] proposed to fit an auto-regressive, moving-average model (ARMA) using the residual data after detrending. This test is designed to show that the data cannot come from a linear stationary process if a large Kendall's $\tau$ value is obtained from it. When this test gives a *p*-value as low as 0.1%, that does not mean that the probability of critical transition is as high as 99.9%. It means that the probability that this time-series data is generated using a linear stationary model is as low as 0.1%. Boettiger & Hastings [21] proposed to fit two nonlinear models that both have a normal form for the saddle-node bifurcation. The difference between these two models is that one of them has a fixed bifurcation parameter, while the other has a changing parameter. They compared the distributions of the test statistics generated from these two models to determine if these models are statistically different and to explore why one of them better describes the data.

Here, we consider an ARMA model as an example of parametric method and evaluate its performance in approximating Kendall's $\tau$ distribution obtained from the harvesting model (equation (2.1)). Similar to the previous example, 200 distinct time series are first generated using the (harvesting) model. Next, we use one of the time series to fit an ARMA model, and generate another distinct 200 time series using the fitted ARMA model. The distribution of Kendall's $\tau$ calculated from the initially generated time series and all the ARMA time series are shown in figure 7. Results show that the distribution approximated by the ARMA model is close to the reference distribution directly obtained by the time series of the harvesting model.

For this example, we compared the results of the performed parametric test using an ARMA model and the non-parametric modified Mann–Kendall test. Using the same single time series selected to generate ARMA results, the non-parametric modified Mann–Kendall distribution was approximated and plotted on the top of the other distributions in figure 7. Figure 7 shows that all three distributions are close to each other. Therefore, both the parametric ARMA test and the non-parametric modified Mann–Kendall can accurately approximate the distribution of Kendall's $\tau$ calculated from time-series data of low-dimensional systems with Gaussian noise. The benefit of the non-parametric Mann–Kendall test is that the distribution of Kendall's $\tau$ can be estimated directly from a single time series, and thus no further simulation or measurements are required, in contrast to the parametric tests. As a result, the non-parametric modified Mann–Kendall test is much faster computationally than the parametric ARMA test.

# 4. Discussion and conclusion

Kendall's $\tau$ is often used to quantify the trend of statistics related to the critical slowing down phenomenon and detect if the system is at risk of an upcoming transition. Due to the probabilistic nature of Kendall's $\tau$ values, however, making any conclusion about the risk of impending transitions based on these values requires a detailed understanding of the statistical significance of the estimated Kendall's $\tau$ from time series.

In this study, we analysed parameters that affect estimated Kendall's $\tau$ values so that researchers consider these factors when using this metric, particularly in the context of forecasting critical transitions. Results of this study provide a guide for interpreting the Kendall's $\tau$ values and identifying when this metric might not provide a reliable prediction and should not be used. It was demonstrated that decreasing the number of observations below a certain limit affects the trend of the early warning signals. In addition, the low number of observations may cause a system with a non-stationary signal (e.g. a time-varying control parameter $c$) appear similar to a system with a fixed parameter, which is undesirable in the study of critical transitions. Moreover, it was shown that a large window size inflates the value of Kendall's $\tau$ calculated from the same time series, resulting in a false alarm if not interpreted correctly. Given that different parameters affect the statistics of approximated Kendall's $\tau$, the current analysis highlights that performing a significance test on the estimated Kendall's $\tau$ coefficient is necessary.

We summarized and compared selected parametric and non-parametric tests to evaluate the significance of Kendall's $\tau$. Particularly, we proposed to use the non-parametric Mann–Kendall test as an efficient test to assess the reliability of the approximated values. It was demonstrated that both the parametric and non-parametric tests yield similar and valid results for a low-dimensional system with Gaussian stochastic excitation. The benefit of the non-parametric Mann–Kendall test, however, is that the distribution of Kendall's $\tau$ can be estimated directly from the time series, and thus no further data or simulations are required, and the computation is much faster than the parametric tests.

Based on this analysis, we suggest a guideline to consider when using Kendall's $\tau$ to study a system subject to critical slowing down. First, a large window size can inflate the value of Kendall's $\tau$ compared with a small window size. We encourage the use of smaller window sizes when there is a large enough amount of data. Second, we demonstrated that the values of Kendall's $\tau$ are sensitive to the number of available observations of the system. For a smaller number of available data, the probability of estimating random and irrelevant values of Kendall's $\tau$ is higher.

Third, we propose that the non-parametric Mann–Kendall test is a reliable and computationally efficient method to evaluate the significance of the estimated Kendall's $\tau$ values revealing if a detected warning signal is a false alarm or not.

Data accessibility. The codes developed based on the methods described in this article are archived in the Dryad Digital Repository: https://doi.org/10.5061/dryad.c59zw3r7z [27].

Authors' contributions. S.C.: conceptualization, data curation, formal analysis, methodology, software, validation, writing—original draft; A.G.: conceptualization, data curation, formal analysis, methodology, software, validation, writing—original draft; B.I.E.: conceptualization, methodology, supervision, writing—review and editing.

All authors gave final approval for publication and agreed to be held accountable for the work performed therein.

Conflict of interest declaration. We declare we have no competing interest.

Funding. This research was supported by the National Institute of General Medical Sciences of the National Institutes of Health under award no. U01GM110744.

Acknowledgements. The content is solely the responsibility of the authors and does not necessarily reflect the official views of the National Institutes of Health.

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
