## [Peer Review File · Royal Society Open Science]

Review History

RSOS-211346.R0 (Original submission)

Review form: Reviewer 1

Is the manuscript scientifically sound in its present form?

Yes

Are the interpretations and conclusions justified by the results?

Yes

Is the language acceptable?

Yes

Do you have any ethical concerns with this paper?

No

Have you any concerns about statistical analyses in this paper?

No

Recommendation?

Accept with minor revision (please list in comments)

Comments to the Author(s)

In this manuscript the authors propose to use the modified Mann-Kendall test to test the significance of a trend in autocorrelation or variance for the analysis of early warning signals. They indicate that a significance test is needed to evaluate these indicators, but that this is challenging due to the autocorrelation.

General comments:

I certainly agree that a special method is needed to assess the significance as the data are not independent. I was not aware of this method and I think this is a very useful addition to the existing methods. I am not a statistician so I cannot fully judge whether the conditions for this test are met. Intuitively it seems to me that there is a difference between autocorrelation in a measured time series with “natural” autocorrelation and a time series where the autocorrelation is partly artificial as it is due to overlapping windows. It seems that the latter case the data are “less independent” but it could be that this modified test can perfectly well deal with this. Maybe the authors can discuss this point.

Figure 4 shows that the distribution is indeed closer to the distribution from repetitive runs of the stochastic model. Maybe the authors can show this for more cases than only one example (the figures could go to an appendix). They could show it also for different windows and number of observations, like figure 2 and 3.

I think the authors should also discuss another method to assess the significance that is often used in the early warning literature, namely the use of surrogate data (see for instance Dakos et al. 2008, see reference below, or Scheffer, et al.. 2021. Loss of resilience preceded transformations of pre-Hispanic Pueblo societies. PNAS 118:e2024397118). So the idea is to generate 100 or more surrogate data sets with the same power spectrum and test whether the Kendall tau of the data is larger than a percentile of the randomized surrogate data (see Ebisuzaki 1997. A method to estimate the statistical significance of a correlation when the data are serially correlated. Journal of Climate 10:2147-2153.). This method is certainly slower than the modified Mann-Kendall test, but it would be interesting for the reader to know whether this is equally correct or maybe less powerful.

Minor points:

The paper Dakos et al. 2008 (Dakos, et al. 2008. Slowing down as an early warning signal for abrupt climate change. PNAS 105:14308-14312.) should be cited as this is the first article that uses the sliding window approach.

Page 2 line 45. A bit confusing as the method should also work when the shift is desirable. So better change this in something like: “Such transition can be undesirable”

Review form: Reviewer 2**Is the manuscript scientifically sound in its present form?**

Yes

Are the interpretations and conclusions justified by the results?

Yes

Is the language acceptable?

Yes

Do you have any ethical concerns with this paper?

No

Have you any concerns about statistical analyses in this paper?

No

Recommendation?

Major revision is needed (please make suggestions in comments)

Comments to the Author(s)

Report on the manuscript

“Practical guide of using Kendall’s τ in the context of forecasting critical transitions”

By Shiyang Chen, Amin Ghadami, Bogdan Epureanu

The authors explore the use of a rank correlation coefficient (Kendall’s τ) in order to predict critical transitions in time series. Indeed, the effect of parameters such as the window size, sample rate and length of the time series on Kendall’s τ is studied. Existing theoretical background on this coefficient is missing. As an illustrating example serves a logistic differential equation with harvesting perturbed by white noise. Concerning this model, it would be illustrating and helpful to obtain more information on the analytical behavior and the bifurcations responsible for critical transitions.

Moreover, both non-parametric tests (Mann-Kendall) and parametric tests are discussed to obtain insights in the significance of Kendall’s τ .

Overall, the manuscript is carefully written and the conclusions appear to be convincing. The arguments are phenomenological and not analytical. They might indicate some trends, however, they are not mathematical.

Typos

Page 2, line 6 write “a phenomenon” instead of “phenomenon”

Review form: Reviewer 3

Is the manuscript scientifically sound in its present form?

Yes

Are the interpretations and conclusions justified by the results?

Yes

Is the language acceptable?

Yes

Do you have any ethical concerns with this paper?

No

Have you any concerns about statistical analyses in this paper?

No

Recommendation?

Accept with minor revision (please list in comments)

Comments to the Author(s)

In this paper, the authors investigate how the performance of Kendall's τ as a measure of early warning signals in a simple SDE model exhibiting critical transitions depends on parameters such as the size of the moving window used to calculate the indicator variable and the observation frequency of the time series data. They show that larger moving windows yield larger-magnitude Kendall's τ values even in a stationary system due to an increased temporal correlation of indicator variables computed, with serious implications for significance testing. This feature of Kendall's τ is of crucial importance for its use in this context but has received comparatively little attention in the literature up to now. The authors propose an alternative to the standard approach of significance testing by fitting a generic stationary parametric model such as ARMA to the time series: a modification of the Mann-Kendall test to account for autocorrelation in the indicator variables that is computationally faster (and perhaps just as importantly, seemingly more straightforward in its implementation) than ARMA, and show that it performs comparably. A final, and slightly less surprising and interesting result, concerns the observation rate: the fewer observations are made the weaker the signals picked up by Kendall's τ . The main weakness of this part of the study is that this is most likely caused by errors in estimating the indicator variables themselves, rather than being related to the use of Kendall's τ itself.

Overall, the conclusions are interesting and have important practical implications for the use of Kendall's τ in the forecasting of critical transitions. The methodology seems sound, and the manuscript is generally well-written, although some improvements need to be made to clarify some of the methodology, as I state below. I recommend that it be accepted for publication once the following comments have been addressed:

General comments

- In section 2.1 you only consider the effect of window size on the distribution of Kendall's τ in stationary systems. Did you investigate nonstationary systems at all? At least a short passage discussing the implications for nonstationary systems would be helpful.
- The fact that the Kendall's τ values generally vary substantially across the individual time series is itself an important point regarding the high likelihood of false positives and negatives and should be emphasised.
- In section 3, it seems like your approach is to compute the expected null distributions for Kendall's τ , but this isn't made clear. This should be explicitly stated and the precise connection of the distributions in Figs 4 and 5 to the process of significance testing should be explained.
- Is there some way to put Figure 5 into context, perhaps by plotting them alongside the distribution of the unmodified Mann-Kendall test or through a verbal explanation? When they are only plotted alongside one another it's hard to find the conclusion that they are 'close to one another' because there are clearly also substantial differences.

Specific comments:

- The title could perhaps be better phrased as 'Practical guide to using Kendall's τ ...'
- Abstract, line 23: 'approaching to an impending transition' -> 'an approach of an impending transition'
- page 2 line 14: the dynamics of a system don't become slower during the approach to a tipping point, instead it's the recovery of the system from perturbations that slow down.

- p4 line 16-19: 'Positive correlation among the observations increases the chance of obtaining a large Kendall's τ , even in the absence of an actual trend' is a very interesting point, and a reference to an older paper explaining this property of Kendall's τ would be very helpful here.
- p4 lines 29-31: could you briefly state the dynamics that are seen in the harvesting model for this parameter – is there a single stable equilibrium, bistability etc.?
- p6 line 14: 'The time series data is then down sampled to obtained another time series data with...' -> 'The time series data is then down sampled to obtain another time series with...'
- p8 line 8: For which values of c ? Is the harvesting model stationary here or not?
- Figure 5: Some statistics concerning the mean, variance, and especially the skew of these distributions could strengthen your claims that they are similar.
- p11 lines 16-19: 'Third, we propose that the significance of the obtained Kendall's τ values should be studied using either the non-parametric Mann-Kendall test or parametric tests.' should probably be reworded to emphasise that the Mann-Kendall test is a viable and possibly more practical alternative to parametric tests.

Decision letter (RSOS-211346.R0)

Dear Dr Ghadami

The Editors assigned to your paper RSOS-211346 "Practical guide of using Kendall's τ in the context of forecasting critical transitions" have now received comments from reviewers and would like you to revise the paper in accordance with the reviewer comments and any comments from the Editors. Please note this decision does not guarantee eventual acceptance.

Please submit your revised manuscript and required files (see below) no later than 21 days from today's (ie 02-Dec-2021) date. Note: the ScholarOne system will 'lock' if submission of the revision is attempted 21 or more days after the deadline. If you do not think you will be able to meet this deadline please contact the editorial office immediately.

on behalf of Prof Mark Chaplain (Subject Editor)
openscience@royalsociety.org

Associate Editor Comments to Author:

A thread that seems to run through the reviewers' reports is that, while the paper likely has merits and represents a contribution to the literature, it is not currently as clear as might be preferred. The reviewers recommend performing a more thorough literature review (among other requirements) to more clearly situate the paper within the existing body of work in this space. When you have revised the paper, please resubmit for consideration.

Reviewer comments to Author:

Reviewer: 1

Comments to the Author(s)

In this manuscript the authors propose to use the modified Mann-Kendall test to test the significance of a trend in autocorrelation or variance for the analysis of early warning signals. They indicate that a significance test is needed to evaluate these indicators, but that this is challenging due to the autocorrelation.

General comments:

I certainly agree that a special method is needed to assess the significance as the data are not independent. I was not aware of this method and I think this is a very useful addition to the existing methods. I am not a statistician so I cannot fully judge whether the conditions for this test are met. Intuitively it seems to me that there is a difference between autocorrelation in a measured time series with "natural" autocorrelation and a time series where the autocorrelation is partly artificial as it is due to overlapping windows. It seems that the latter case the data are "less independent" but it could be that this modified test can perfectly well deal with this. Maybe the authors can discuss this point.

Figure 4 shows that the distribution is indeed closer to the distribution from repetitive runs of the stochastic model. Maybe the authors can show this for more cases than only one example (the figures could go to an appendix). They could show it also for different windows and number of observations, like figure 2 and 3.

I think the authors should also discuss another method to assess the significance that is often used in the early warning literature, namely the use of surrogate data (see for instance Dakos et al. 2008, see reference below, or Scheffer, et al.. 2021. Loss of resilience preceded transformations of pre-Hispanic Pueblo societies. PNAS 118:e2024397118). So the idea is to generate 100 or more surrogate data sets with the same power spectrum and test whether the Kendall tau of the data is larger than a percentile of the randomized surrogate data (see Ebisuzaki 1997. A method to estimate the statistical significance of a correlation when the data are serially correlated. Journal of Climate 10:2147-2153.). This method is certainly slower than the modified Mann-Kendall test, but it would be interesting for the reader to know whether this is equally correct or maybe less powerful.

Minor points:

The paper Dakos et al. 2008 (Dakos, et al. 2008. Slowing down as an early warning signal for abrupt climate change. PNAS 105:14308–14312.) should be cited as this is the first article that uses the sliding window approach.

Page 2 line 45. A bit confusing as the method should also work when the shift is desirable. So better change this in something like: “Such transition can be undesirable”

Reviewer: 2

Comments to the Author(s)

Report on the manuscript

“Practical guide of using Kendall’s τ in the context of forecasting critical transitions”

By Shiyang Chen, Amin Ghadami, Bogdan Epureanu

The authors explore the use of a rank correlation coefficient (Kendall’s τ) in order to predict critical transitions in time series. Indeed, the effect of parameters such as the window size, sample rate and length of the time series on Kendall’s τ is studied. Existing theoretical background on this coefficient is missing. As an illustrating example serves a logistic differential equation with harvesting perturbed by white noise. Concerning this model, it would be illustrating and helpful to obtain more information on the analytical behavior and the bifurcations responsible for critical transitions.

Moreover, both non-parametric tests (Mann-Kendall) and parametric tests are discussed to obtain insights in the significance of Kendall’s τ .

Overall, the manuscript is carefully written and the conclusions appear to be convincing. The arguments are phenomenological and not analytical. They might indicate some trends, however, they are not mathematical.

Typos

Page 2, line 6 write “a phenomenon” instead of “phenomenon”

Reviewer: 3

Comments to the Author(s)

In this paper, the authors investigate how the performance of Kendall’s τ as a measure of early warning signals in a simple SDE model exhibiting critical transitions depends on parameters such as the size of the moving window used to calculate the indicator variable and the observation frequency of the time series data. They show that larger moving windows yield larger-magnitude Kendall’s τ values even in a stationary system due to an increased temporal correlation of indicator variables computed, with serious implications for significance testing. This feature of Kendall’s τ is of crucial importance for its use in this context but has received comparatively little attention in the literature up to now. The authors propose an alternative to the standard approach of significance testing by fitting a generic stationary parametric model such as ARMA to the time series: a modification of the Mann-Kendall test to account for autocorrelation in the indicator variables that is computationally faster (and perhaps just as importantly, seemingly more straightforward in its implementation) than ARMA, and show that it performs comparably. A final, and slightly less surprising and interesting result, concerns the observation rate: the fewer observations are made the weaker the signals picked up by Kendall’s τ . The main

weakness of this part of the study is that this is most likely caused by errors in estimating the indicator variables themselves, rather than being related to the use of Kendall's tau itself.

Overall, the conclusions are interesting and have important practical implications for the use of Kendall's tau in the forecasting of critical transitions. The methodology seems sound, and the manuscript is generally well-written, although some improvements need to be made to clarify some of the methodology, as I state below. I recommend that it be accepted for publication once the following comments have been addressed:

General comments

- In section 2.1 you only consider the effect of window size on the distribution of Kendall's tau in stationary systems. Did you investigate nonstationary systems at all? At least a short passage discussing the implications for nonstationary systems would be helpful.
- The fact that the Kendall's tau values generally vary substantially across the individual time series is itself an important point regarding the high likelihood of false positives and negatives and should be emphasised.
- In section 3, it seems like your approach is to compute the expected null distributions for Kendall's tau, but this isn't made clear. This should be explicitly stated and the precise connection of the distributions in Figs 4 and 5 to the process of significance testing should be explained.
- Is there some way to put Figure 5 into context, perhaps by plotting them alongside the distribution of the unmodified Mann-Kendall test or through a verbal explanation? When they are only plotted alongside one another it's hard to find the conclusion that they are 'close to one another' because there are clearly also substantial differences.

Specific comments:

- The title could perhaps be better phrased as 'Practical guide to using Kendall's τ ...'
- Abstract, line 23: 'approaching to an impending transition' -> 'an approach of an impending transition'
- page 2 line 14: the dynamics of a system don't become slower during the approach to a tipping point, instead it's the recovery of the system from perturbations that slow down.
- p4 line 16-19: 'Positive correlation among the observations increases the chance of obtaining a large Kendall's τ , even in the absence of an actual trend' is a very interesting point, and a reference to an older paper explaining this property of Kendall's τ would be very helpful here.
- p4 lines 29-31: could you briefly state the dynamics that are seen in the harvesting model for this parameter - is there a single stable equilibrium, bistability etc.?
- p6 line 14: 'The time series data is then down sampled to obtained another time series data with...' -> 'The time series data is then down sampled to obtain another time series with...'
- p8 line 8: For which values of c ? Is the harvesting model stationary here or not?
- Figure 5: Some statistics concerning the mean, variance, and especially the skew of these distributions could strengthen your claims that they are similar.
- p11 lines 16-19: 'Third, we propose that the significance of the obtained Kendall's τ values should be studied using either the non-parametric Mann-Kendall test or parametric tests.' should probably be reworded to emphasise that the Mann-Kendall test is a viable and possibly more practical alternative to parametric tests.

===PREPARING YOUR MANUSCRIPT===

a 'clean' version of the new manuscript that incorporates the changes made, but does not highlight them. This version will be used for typesetting if your manuscript is accepted. Please ensure that any equations included in the paper are editable text and not embedded images.

If you have been asked to revise the written English in your submission as a condition of publication, you must do so, and you are expected to provide evidence that you have received language editing support. The journal would prefer that you use a professional language editing service and provide a certificate of editing, but a signed letter from a colleague who is a fluent speaker of English is acceptable. Note the journal has arranged a number of discounts for authors using professional language editing services (<https://royalsociety.org/journals/authors/benefits/language-editing/>).

===PREPARING YOUR REVISION IN SCHOLARONE===

- If you are requesting a discretionary waiver for the article processing charge, the waiver form must be included at this step.
- If you are providing image files for potential cover images, please upload these at this step, and inform the editorial office you have done so. You must hold the copyright to any image provided.
- A copy of your point-by-point response to referees and Editors. This will expedite the preparation of your proof.

- Ensure that your data access statement meets the requirements at <https://royalsociety.org/journals/authors/author-guidelines/#data>. You should ensure that you cite the dataset in your reference list. If you have deposited data etc in the Dryad repository, please include both the 'For publication' link and 'For review' link at this stage.
- If you are requesting an article processing charge waiver, you must select the relevant waiver option (if requesting a discretionary waiver, the form should have been uploaded at Step 3 'File upload' above).
- If you have uploaded ESM files, please ensure you follow the guidance at <https://royalsociety.org/journals/authors/author-guidelines/#supplementary-material> to include a suitable title and informative caption. An example of appropriate titling and captioning may be found at https://figshare.com/articles/Table_S2_from_Is_there_a_trade-off_between_peak_performance_and_performance_breadth_across_temperatures_for_aerobic_scope_in_teleost_fishes_/3843624.

Author's Response to Decision Letter for (RSOS-211346.R0)

See Appendix A.

RSOS-211346.R1 (Revision)

Review form: Reviewer 1

Is the manuscript scientifically sound in its present form?

No

Are the interpretations and conclusions justified by the results?

No

Is the language acceptable?

Yes

Do you have any ethical concerns with this paper?

No

Have you any concerns about statistical analyses in this paper?

Yes

Recommendation?

Major revision is needed (please make suggestions in comments)

Comments to the Author(s)

In this revised manuscript the authors propose to use the modified Mann-Kendall test for testing if a trend in an early warning signal is significant. I was rather positive, but had one important doubt, namely whether the modified test can be used in overlapping windows. Maybe I was not clear enough in my comments, but I think the authors have not dealt with this comment. They replied that indeed this is a problem for the normal Mann-Kendall test. Moreover, I suggested to compare the results with the null model method (Ebisuzaki). (In this method we generate surrogate data using a Fourier transformation with randomized phases. If the real data performs better than the surrogate data, we have a significant effect.)

The authors did reply that they did not test it but 'believe' that the null model will give comparable results. I found that unsatisfying and did myself a fast test to convince myself that the sliding window results of the modified Mann-Kendall test and the null model are the same (and also because I would be interested to use the modified test in future). Unfortunately I found that the modified Mann Kendall test gives much lower p values (i.e. more significant results) than the null model (especially if we use large overlapping windows). I think this must have been caused by pseudo replication due to the sliding window. Therefore I think this test cannot be used on overlapping windows. The authors could test if it works on non-overlapping windows, but probably you will then have too little data.

Review form: Reviewer 4**Is the manuscript scientifically sound in its present form?**

Yes

Are the interpretations and conclusions justified by the results?

Yes

Is the language acceptable?

Yes

Do you have any ethical concerns with this paper?

No

Have you any concerns about statistical analyses in this paper?

No

Recommendation?

Accept as is

Comments to the Author(s)

The paper is a reasonable explanation of potential problem with Kendall's tau for rolling window statistics. Kendall's tau has been used in the theoretical literature on early warnings of critical transitions.

However I am not aware that Kendall's tau is used by ecologists conducting whole-ecosystem field experiments. Real-world science on early warnings has used MAR models or their Bayes' analogs (e.g. Ives & Dakos, Taranu et al.), tests of heteroskedasticity (Seekell), quickest detection (Carpenter), and qualitative comparisons for example. Some papers conducted their own system-specific studies of the effects of window size. The paper provides a useful warning to ecologists as well as theoreticians but perhaps theoreticians are the primary readership.

Carpenter, S. R., Brock, W. A., Cole, J. J., & Pace, M. L. (2014). A new approach for rapid detection of nearby thresholds in ecosystem time series. *Oikos*, 123(3), 290-297. doi:10.1111/j.1600-0706.2013.00539.x

Ives, A. R., & Dakos, V. (2012). Detecting dynamical changes in nonlinear time series using locally linear state-space models. *Ecosphere*, 3(6), art58. doi:10.1890/es11-00347.1

Seekell, D., Carpenter, S., Cline, T., & Pace, M. (2012). Conditional Heteroskedasticity Forecasts Regime Shift in a Whole-Ecosystem Experiment. *Ecosystems*, 15(5), 741-747. doi:10.1007/s10021-012-9542-2

Taranu, Z. E., Carpenter, S. R., Frossard, V., Jenny, J.-P., Thomas, Z., Vermaire, J. C., & Perga, M.-E. (2018). Can we detect ecosystem critical transitions and signals of changing resilience from paleo-ecological records? *Ecosphere*, 9(10), e02438. doi:doi:10.1002/ecs2.2438

Review form: Reviewer 5

Is the manuscript scientifically sound in its present form?

No

Are the interpretations and conclusions justified by the results?

Yes

Is the language acceptable?

Yes

Do you have any ethical concerns with this paper?

No

Have you any concerns about statistical analyses in this paper?

No

Recommendation?

Major revision is needed (please make suggestions in comments)

Comments to the Author(s)

The manuscript by Chen et al. discusses the utility of Kendall's tau for assessing trends in indicators of critical slowing down, the importance of a significance test for the Kendall's tau estimate, and a non-parametric solution for that significance test. The topic is methodologically valuable and the authors explore the influence of various aspects of the data on the statistical estimates and tests. The exploration of moving window size and amount of data available are valuable. The authors have improved the text in response to the previous reviewers comments. There are a few additional points that can be improved upon.

I did not find the description of the moving window implementation in 2.1 complete. Was the window moved one observation at a time? Figure 1 seems to indicate movement of half the window size. This seems to be a key aspect of the resulting autocorrelation and not unambiguous. Further, do the authors mean the window sizes here as percentages of the total time series (implied by P7 reference to a window of 50% of the time series, but not actually stated that way in earlier text or figures)? Finally, what is the relationship between window size and amount of data available, in other words explain to the reader why the statement on P13 L23 holds: “We encourage the use of smaller window sizes when there is a large enough amount of data.” What is the standard practice and suggestion for window sizes and overlap?

The authors find that for their model of choice a time series of length <200 “the distribution becomes almost symmetric about zero, which is associated normally with random signals”. If harvesting is measured on an annual time scale this is an unlikely amount of data to have. What does this mean for the power of Kendall’s tau in many of the ecological regime shifts under consideration here? Or if this is purely a function of the other parameters in the model that authors should include some discussion on this point. It is referred to again at the end with a suggestion not to detrend, but I don’t think this was explained/brought up prior to the conclusion.

On page 11 the authors mention 2 parametric tests and say that they are going to compare them (L24-28), but the only test compared are ARMA with the modified Mann-Kendall.

Clearly the modified Mann-Kendall is more straightforward and efficient. The authors mention in the introduction that parametric tests can be more powerful. Does Figure 6 suggest similar power in this scenario or can the authors speak to that point?

The conclusion summarizes the findings on Kendall’s tau. Nowhere in the paper are alternatives to Kendall’s tau mentioned or whether the issues raised here suggest cases when Kendall’s tau should not be used.

Minor points:

I found the text label “Trend: Kendall’s tau” on Figure 1 to be confusing as it seemed to correspond to the addition of the single new measurement, rather than a calculation across all the indicator values.

P2 L13 Check verb tense agreement

P3 L15 Check verb number agreement

P5 L53 “parameter” -> “parameters”

P7 L48 Check verb tense

P11 L26 Missing word?

Decision letter (RSOS-211346.R1)

Dear Dr Ghadami

The Editors assigned to your paper RSOS-211346.R1 "Practical guide to using Kendall's τ in the context of forecasting critical transitions" have now received comments from reviewers and would like you to revise the paper in accordance with the reviewer comments and any comments from the Editors. Please note this decision does not guarantee eventual acceptance.

Please submit your revised manuscript and required files (see below) no later than 21 days from today's (ie 31-Mar-2022) date. Note: the ScholarOne system will 'lock' if submission of the revision is attempted 21 or more days after the deadline. If you do not think you will be able to meet this deadline please contact the editorial office immediately.

on behalf of Prof Mark Chaplain (Subject Editor)
openscience@royalsociety.org

Associate Editor Comments to Author:

Dear authors,

Firstly, our apologies for the delay in returning a decision on your revised paper. Unfortunately, one of the original referees was not available to review the second version of your paper. As such, we secured a new third referee who highlights similar concerns as noted by Reviewer #1 in terms of window sizes and overlap. We do not usually allow multiple rounds of revision, but on this occasion we hope that one final round of major revisions will help resolve these issues. Please note that we cannot guarantee publication until these issues are addressed. If you require any additional time to resubmit, please let the editorial office know.

Best wishes
Reviewer comments to Author:
Reviewer: 1
Comments to the Author(s)

In this revised manuscript the authors propose to use the modified Mann-Kendall test for testing if a trend in an early warning signal is significant. I was rather positive, but had one important doubt, namely whether the modified test can be used in overlapping windows. Maybe I was not clear enough in my comments, but I think the authors have not dealt with this comment. They replied that indeed this is a problem for the normal Mann-Kendall test. Moreover, I suggested to compare the results with the null model method (Ebisuzaki). (In this method we generate surrogate data using a Fourier transformation with randomized phases. If the real data performs better than the surrogate data, we have a significant effect.)

The authors did reply that they did not test it but 'believe' that the null model will give comparable results. I found that unsatisfying and did myself a fast test to convince myself that the sliding window results of the modified Mann-Kendall test and the null model are the same (and also because I would be interested to use the modified test in future). Unfortunately I found that the modified Mann Kendall test gives much lower p values (i.e. more significant results) than the null model (especially if we use large overlapping windows). I think this must have been caused by pseudo replication due to the sliding window. Therefore I think this test cannot be used on overlapping windows. The authors could test if it works on non-overlapping windows, but probably you will then have too little data.

Reviewer: 4

Comments to the Author(s)

The paper is a reasonable explanation of potential problem with Kendall's tau for rolling window statistics. Kendall's tau has been used in the theoretical literature on early warnings of critical transitions.

However I am not aware that Kendall's tau is used by ecologists conducting whole-ecosystem field experiments. Real-world science on early warnings has used MAR models or their Bayes' analogs (e.g. Ives & Dakos, Taranu et al.), tests of heteroskedasticity (Seekell), quickest detection (Carpenter), and qualitative comparisons for example. Some papers conducted their own system-specific studies of the effects of window size. The paper provides a useful warning to ecologists as well as theoreticians but perhaps theoreticians are the primary readership.

Carpenter, S. R., Brock, W. A., Cole, J. J., & Pace, M. L. (2014). A new approach for rapid detection of nearby thresholds in ecosystem time series. *Oikos*, 123(3), 290-297. doi:10.1111/j.1600-0706.2013.00539.x

Ives, A. R., & Dakos, V. (2012). Detecting dynamical changes in nonlinear time series using locally linear state-space models. *Ecosphere*, 3(6), art58. doi:10.1890/es11-00347.1

Seekell, D., Carpenter, S., Cline, T., & Pace, M. (2012). Conditional Heteroskedasticity Forecasts Regime Shift in a Whole-Ecosystem Experiment. *Ecosystems*, 15(5), 741-747. doi:10.1007/s10021-012-9542-2

Taranu, Z. E., Carpenter, S. R., Frossard, V., Jenny, J.-P., Thomas, Z., Vermaire, J. C., & Perga, M.-E. (2018). Can we detect ecosystem critical transitions and signals of changing resilience from paleo-ecological records? *Ecosphere*, 9(10), e02438. doi:doi:10.1002/ecs2.2438

Reviewer: 5

Comments to the Author(s)

The manuscript by Chen et al. discusses the utility of Kendall's tau for assessing trends in indicators of critical slowing down, the importance of a significance test for the Kendall's tau

estimate, and a non-parametric solution for that significance test. The topic is methodologically valuable and the authors explore the influence of various aspects of the data on the statistical estimates and tests. The exploration of moving window size and amount of data available are valuable. The authors have improved the text in response to the previous reviewers comments. There are a few additional points that can be improved upon.

I did not find the description of the moving window implementation in 2.1 complete. Was the window moved one observation at a time? Figure 1 seems to indicate movement of half the window size. This seems to be a key aspect of the resulting autocorrelation and not unambiguous. Further, do the authors mean the window sizes here as percentages of the total time series (implied by P7 reference to a window of 50% of the time series, but not actually stated that way in earlier text or figures)? Finally, what is the relationship between window size and amount of data available, in other words explain to the reader why the statement on P13 L23 holds: "We encourage the use of smaller window sizes when there is a large enough amount of data." What is the standard practice and suggestion for window sizes and overlap?

The authors find that for their model of choice a time series of length <200 "the distribution becomes almost symmetric about zero, which is associated normally with random signals". If harvesting is measured on an annual time scale this is an unlikely amount of data to have. What does this mean for the power of Kendall's tau in many of the ecological regime shifts under consideration here? Or if this is purely a function of the other parameters in the model that authors should include some discussion on this point. It is referred to again at the end with a suggestion not to detrend, but I don't think this was explained/brought up prior to the conclusion.

On page 11 the authors mention 2 parametric tests and say that they are going to compare them (L24-28), but the only test compared are ARMA with the modified Mann-Kendall.

Clearly the modified Mann-Kendall is more straightforward and efficient. The authors mention in the introduction that parametric tests can be more powerful. Does Figure 6 suggest similar power in this scenario or can the authors speak to that point?

The conclusion summarizes the findings on Kendall's tau. Nowhere in the paper are alternatives to Kendall's tau mentioned or whether the issues raised here suggest cases when Kendall's tau should not be used.

Minor points:

I found the text label "Trend: Kendall's tau" on Figure 1 to be confusing as it seemed to correspond to the addition of the single new measurement, rather than a calculation across all the indicator values.

P2 L13 Check verb tense agreement

P3 L15 Check verb number agreement

P5 L53 "parameter" -> "parameters"

P7 L48 Check verb tense

P11 L26 Missing word?

===PREPARING YOUR MANUSCRIPT===

one version identifying all the changes that have been made (for instance, in coloured highlight, in bold text, or tracked changes);
 a 'clean' version of the new manuscript that incorporates the changes made, but does not highlight them. This version will be used for typesetting if your manuscript is accepted.

If you have been asked to revise the written English in your submission as a condition of publication, you must do so, and you are expected to provide evidence that you have received language editing support. The journal would prefer that you use a professional language editing service and provide a certificate of editing, but a signed letter from a colleague who is a fluent speaker of English is acceptable. Note the journal has arranged a number of discounts for authors using professional language editing services (<https://royalsociety.org/journals/authors/benefits/language-editing/>).

===PREPARING YOUR REVISION IN SCHOLARONE===

- Any electronic supplementary material (ESM).
- If you are requesting a discretionary waiver for the article processing charge, the waiver form must be included at this step.
- If you are providing image files for potential cover images, please upload these at this step, and inform the editorial office you have done so. You must hold the copyright to any image provided.
- A copy of your point-by-point response to referees and Editors. This will expedite the preparation of your proof.

- Ensure that your data access statement meets the requirements at <https://royalsociety.org/journals/authors/author-guidelines/#data>. You should ensure that you cite the dataset in your reference list. If you have deposited data etc in the Dryad repository, please include both the 'For publication' link and 'For review' link at this stage.
- If you are requesting an article processing charge waiver, you must select the relevant waiver option (if requesting a discretionary waiver, the form should have been uploaded at Step 3 'File upload' above).
- If you have uploaded ESM files, please ensure you follow the guidance at <https://royalsociety.org/journals/authors/author-guidelines/#supplementary-material> to include a suitable title and informative caption. An example of appropriate titling and captioning may be found at https://figshare.com/articles/Table_S2_from_Is_there_a_trade-off_between_peak_performance_and_performance_breadth_across_temperatures_for_aerobic_scope_in_teleost_fishes_/3843624.

Author's Response to Decision Letter for (RSOS-211346.R1)

See Appendix B.

RSOS-211346.R2 (Revision)

Review form: Reviewer 5

Is the manuscript scientifically sound in its present form?

Yes

Are the interpretations and conclusions justified by the results?

Yes

Is the language acceptable?

Yes

Do you have any ethical concerns with this paper?

No

Have you any concerns about statistical analyses in this paper?

No

Recommendation?

Accept with minor revision (please list in comments)

Comments to the Author(s)

The authors have done a nice job of responding to reviewer comments, including expanding the consideration of window sizes and overlaps. I found the new version improved and clear. I have only a couple small items I noticed:

P3 L21 Awkward sentence structure in new text

P6 L40 and Figure 2 caption: the text says "smaller window size and a large window size". The panels are in order of largest to smallest. Would be better to match these in one order or the other.

P10 L18 Is it necessary/helpful to say anything about calculating the autocorrelation? Does this reduce the degrees of freedom in the hypothesis test?

P10 L34 Is it meant to say Eq 4?

Decision letter (RSOS-211346.R2)

Dear Dr Ghadami,

On behalf of the Editors, we are pleased to inform you that your Manuscript RSOS-211346.R2 "Practical guide to using Kendall's τ in the context of forecasting critical transitions" has been accepted for publication in Royal Society Open Science subject to minor revision in accordance with the referees' reports. Please find the referees' comments along with any feedback from the Editors below my signature.

Please submit your revised manuscript and required files (see below) no later than 7 days from today's (ie 20-Jun-2022) date. Note: the ScholarOne system will 'lock' if submission of the revision is attempted 7 or more days after the deadline. If you do not think you will be able to meet this deadline please contact the editorial office immediately.

Kind regards,

Royal Society Open Science Editorial Office
Royal Society Open Science

on behalf of Professor Mark Chaplain (Subject Editor)
openscience@royalsociety.org

Reviewer comments to Author:

Reviewer: 5

Comments to the Author(s)

The authors have done a nice job of responding to reviewer comments, including expanding the consideration of window sizes and overlaps. I found the new version improved and clear. I have only a couple small items I noticed:

P3 L21 Awkward sentence structure in new text

P6 L40 and Figure 2 caption: the text says "smaller window size and a large window size". The panels are in order of largest to smallest. Would be better to match these in one order or the other.

P10 L18 Is it necessary/helpful to say anything about calculating the autocorrelation? Does this reduce the degrees of freedom in the hypothesis test?

P10 L34 Is it meant to say Eq 4?

===PREPARING YOUR MANUSCRIPT===

one version should clearly identify all the changes that have been made (for instance, in coloured highlight, in bold text, or tracked changes);

===PREPARING YOUR REVISION IN SCHOLARONE===

-- If you are requesting an article processing charge waiver, you must select the relevant waiver option (if requesting a discretionary waiver, the form should have been uploaded, see 'File upload' above).

-- If you have uploaded any electronic supplementary (ESM) files, please ensure you follow the guidance at <https://royalsociety.org/journals/authors/author-guidelines/#supplementary-material> to include a suitable title and informative caption. An example of appropriate titling and captioning may be found at https://figshare.com/articles/Table_S2_from_Is_there_a_trade-

off_between_peak_performance_and_performance_breadth_across_temperatures_for_aerobic_sc
ope_in_teleost_fishes_/3843624.

Author's Response to Decision Letter for (RSOS-211346.R2)

See Appendix C.

Decision letter (RSOS-211346.R3)

Dear Dr Ghadami:

I am pleased to inform you that your manuscript entitled "Practical guide to using Kendall's τ in the context of forecasting critical transitions" is now accepted for publication in Royal Society Open Science.

Please remember to make any data sets or code libraries 'live' prior to publication, and update any links as needed when you receive a proof to check - for instance, from a private 'for review' URL to a publicly accessible 'for publication' URL. It is also good practice to add data sets, code and other digital materials to your reference list.

Royal Society Open Science is a fully open access journal. A payment may be due before your article is published. Our partner Copyright Clearance Center's RightsLink for Scientific Communications will contact the corresponding author about your open access options from the email domain @copyright.com (if you have any queries regarding fees, please see <https://royalsocietypublishing.org/rsos/charges> or contact authorfees@royalsociety.org).

Please see the Royal Society Publishing guidance on how you may share your accepted author manuscript at <https://royalsociety.org/journals/ethics-policies/media-embargo/>. After publication, some additional ways to effectively promote your article can also be found here

<https://royalsociety.org/blog/2020/07/promoting-your-latest-paper-and-tracking-your-results/>.

on behalf of Professor Mark Chaplain (Subject Editor).

Follow Royal Society Publishing on Twitter: @RSocPublishing
Follow Royal Society Publishing on Facebook:
<https://www.facebook.com/RoyalSocietyPublishing/>
Read Royal Society Publishing's blog:
<https://royalsociety.org/blog/blogsearchpage/?category=Publishing>

Appendix A

To: **Andrew Dunn**
Senior Publishing Editor, Journal of the Royal Society Open Science
Re: Manuscript ID: RSOS-211346

January 25, 2022

Dear Dr. Dunn,

We would like to thank you and the reviewers for the feedback. The comments of the reviewers have been very helpful in improving our paper in terms of content and readability, for which we are deeply grateful. We have revised our paper and addressed the reviewers' comments. In particular, below are details regarding some of our specific corrections and revisions we made (*shown in blue*) in response to reviewers' comments.

Sincerely,
the authors

Reviewer: 1

Comments to the Author(s)

In this manuscript the authors propose to use the modified Mann-Kendall test to test the significance of a trend in autocorrelation or variance for the analysis of early warning signals. They indicate that a significance test is needed to evaluate these indicators, but that this is challenging due to the autocorrelation.

General comments:

I certainly agree that a special method is needed to assess the significance as the data are not independent. I was not aware of this method and I think this is a very useful addition to the existing methods. I am not a statistician so I cannot fully judge whether the conditions for this test are met. Intuitively it seems to me that there is a difference between autocorrelation in a measured time series with “natural” autocorrelation and a time series where the autocorrelation is partly artificial as it is due to overlapping windows. It seems that the latter case the data are “less independent” but it could be that this modified test can perfectly well deal with this. Maybe the authors can discuss this point.

Thank you for this comment. As the reviewer correctly points out, serial dependence in the data can affect the statistics approximated using data. The null hypothesis in the Mann-Kendall test is that the data are independent, randomly ordered, without serial correlation structure among observations. However, in many real situations the observed data can be autocorrelated, which can result in misinterpretation of trend test results. Such an autocorrelation can come from any sources, e.g. seasonality in measured environmental data or situations similar to our study where postprocessing of the data causes such an autocorrelation. We discussed this point in more details in the revised text. Specifically, we noted in Section 3.1. that:

“The non-parametric Mann-Kendall test is commonly employed to detect monotonic trends in time series. The null hypothesis for the traditional Mann-Kendall trend test, however, is that the data are independent, randomly ordered, without serial correlation structure among the observations. However, in many real situations the observed data are autocorrelated, which can result in misinterpretation of trend test results. In particular, in the context of early warning signals, this null hypothesis is not rigorously obeyed for the series of statistics such as the standard deviation or the autocorrelation that are obtained from a time series using a moving window. As a result, using the standard Mann-Kendall test does not lead to reliable results in identifying the significance of the trend of the early warning signals.”

This is also briefly highlighted throughout the revised text, including the introduction and the first paragraph of section 2.1.

Figure 4 shows that the distribution is indeed closer to the distribution from repetitive runs of the stochastic model. Maybe the authors can show this for more cases than only one example (the figures could go to an appendix). They could show it also for different windows and number of observations, like figure 2 and 3.

We included more cases to show the effects of changes in window size and number of observations on the results. The revised figure shows that changes in the window size have a significant effect on the distribution of Kendall’s τ , while changes in the observation frequency do not affect the

distribution in the case of stationary signals studied in this example. However, note that as discussed in Section 2.2., changing the number of observations may cause a system with a non-stationary signal (varying control parameter c) appear similar to a system with a fixed parameter, which should be avoided in the study of critical transitions. As a result, measuring enough data and performing a significance test on the measured Kendall's τ values are both necessary when studying critical transitions. We clarified this in the conclusions.

I think the authors should also discuss another method to assess the significance that is often used in the early warning literature, namely the use of surrogate data (see for instance Dakos et al. 2008, see reference below, or Scheffer, et al. 2021. Loss of resilience preceded transformations of pre-Hispanic Pueblo societies. PNAS 118:e2024397118). So the idea is to generate 100 or more surrogate data sets with the same power spectrum and test whether the Kendall tau of the data is larger than a percentile of the randomized surrogate data (see Ebisuzaki 1997. A method to estimate the statistical significance of a correlation when the data are serially correlated. Journal of Climate 10:2147-2153.). This method is certainly slower than the modified Mann-Kendall test, but it would be interesting for the reader to know whether this is equally correct or maybe less powerful.

Indeed, there are other approaches in the literature, including the ones suggested by the reviewer, that can be used to test the significance of approximated statistics from time series. As the reviewer correctly points out, these methods usually rely on generating a large number of surrogate data and evaluate the significance based on the randomized surrogate data. In this study, in addition to the modified Mann-Kendall test, we used a parametric test, namely the auto-regressive moving-average model (ARMA), suggested by Dakos et al. [8] to compare the results with those obtained by the modified Mann-Kendall test. The results show that the approximated distributions obtained by the ARMA method are similar to those of the modified Mann-Kendall method. However, as the reviewer correctly points out and we also highlighted in the text, the modified Mann-Kendall test is much faster since it does not require additional surrogate simulations. We believe that the other approaches that the reviewer suggested will also result in similar conclusions.

Minor points:

The paper Dakos et al. 2008 (Dakos, et al. 2008. Slowing down as an early warning signal for abrupt climate change. PNAS 105:14308–14312.) should be cited as this is the first article that uses the sliding window approach.

We cited this paper in the revised text as reference 13. Thank you for pointing this out.

Page 2 line 45. A bit confusing as the method should also work when the shift is desirable. So better change this in something like: "Such transition can be undesirable"

Thank you for this suggestion. We made this change in the revised manuscript.

Reviewer: 2

Comments to the Author(s)

The authors explore the use of a rank correlation coefficient (Kendall's τ) in order to predict critical transitions in time series. Indeed, the effect of parameters such as the window size, sample rate and length of the time series on Kendall's τ is studied. Existing theoretical background on this coefficient is missing.

Thank you for this suggestion. We added more theoretical background about Kendall's τ coefficient to the revised text in Section 2.

As an illustrating example serves a logistic differential equation with harvesting perturbed by white noise. Concerning this model, it would be illustrating and helpful to obtain more information on the analytical behavior and the bifurcations responsible for critical transitions.

We added more details about the dynamics of the harvesting model and critical transition in this system in Section 2 of the revised text. In addition, we added the bifurcation diagram for this system (Figure 1, revised text), which shows the equilibria and critical transition in this system.

Moreover, both non-parametric tests (Mann-Kendall) and parametric tests are discussed to obtain insights in the significance of Kendall's τ . Overall, the manuscript is carefully written and the conclusions appear to be convincing. The arguments are phenomenological and not analytical. They might indicate some trends, however, they are not mathematical.

Typos

Page 2, line 6 write "a phenomenon" instead of "phenomenon"

Thank you for pointing out this typo. We have corrected it.

Reviewer: 3

Comments to the Author(s)

In this paper, the authors investigate how the performance of Kendall's τ as a measure of early warning signals in a simple SDE model exhibiting critical transitions depends on parameters such as the size of the moving window used to calculate the indicator variable and the observation frequency of the time series data. They show that larger moving windows yield larger-magnitude Kendall's tau values even in a stationary system due to an increased temporal correlation of indicator variables computed, with serious implications for significance testing. This feature of Kendall's tau is of crucial importance for its use in this context but has received comparatively little attention in the literature up to now. The authors propose an alternative to the standard approach of significance testing by fitting a generic stationary parametric model such as ARMA to the time series: a modification of the Mann-Kendall test to account for autocorrelation in the indicator variables that is computationally faster (and perhaps just as importantly, seemingly more straightforward in its implementation) than ARMA, and show that it performs comparably. A final, and slightly less surprising and interesting result, concerns the observation rate: the fewer observations are made the weaker the signals picked up by Kendall's tau. The main weakness of this part of the study is that this is most likely caused by errors in estimating the indicator variables themselves, rather than being related to the use of Kendall's tau itself.

Overall, the conclusions are interesting and have important practical implications for the use of Kendall's tau in the forecasting of critical transitions. The methodology seems sound, and the manuscript is generally well-written, although some improvements need to be made to clarify some of the methodology, as I state below. I recommend that it be accepted for publication once the following comments have been addressed:

General comments

- In section 2.1 you only consider the effect of window size on the distribution of Kendall's tau in stationary systems. Did you investigate nonstationary systems at all? At least a short passage discussing the implications for nonstationary systems would be helpful.

The results shown in Section 2.1 are based on stationary simulations that allow us to directly compare the performance of the modified Mann-Kendall test with histograms obtained from data. Similar results are obtained for non-stationary simulations, except that the histogram leans toward the right indicating a trend in the approximated Kendall's tau values. In the case of non-stationary simulations, the Mann-Kendall test still approximates a stationary distribution as a null hypothesis, which cannot be directly compared with the distribution generated from non-stationary data. Hence, to be able to showcase the accuracy of the Mann-Kendall test approximations compared to the reference distribution, we focused on the stationary simulations in this section.

- The fact that the Kendall's tau values generally vary substantially across the individual time series is itself an important point regarding the high likelihood of false positives and negatives and should be emphasised.

We added a clarification at the end of the revised Section 2.1 to emphasize this issue. In particular, we added the following text to the revised manuscript:

“In addition, one should note that for data recorded under the same parameter condition, Kendall's τ can take a wide range of values, both positive and negative, which might result in a false positive or a false negative alarm. Thus, merely calculating Kendall's τ values is not enough to decide the probability of a critical transition, and a hypothesis test is necessary. A hypothesis test based on modified Mann-Kendall approach is introduced in Section 3.”

- In section 3, it seems like your approach is to compute the expected null distributions for Kendall's tau, but this isn't made clear. This should be explicitly stated and the precise connection of the distributions in Figs 4 and 5 to the process of significance testing should be explained.

Thank you for this comment. We added a brief description about the null hypothesis and the idea behind the presented approaches in section 3. In particular, we noted in the revised text that:

“Because Kendall's τ is affected by several parameters, as discussed in the previous section, merely observing the values of Kendall's τ does not always reveal the desired information about the system. One should evaluate the significance of the approximated Kendall's τ values to identify whether the system is at risk of critical transitions. A number of tests have been proposed to understand the significance of a certain value of Kendall's τ in the literature. These approaches can be categorized as parametric and non-parametric tests. In these approaches, the goal is to identify the null hypothesis that there is no trend in the data, and identify the significance of the approximated Kendall's τ assuming the null hypothesis is true.”

We clarified this point in other sections throughout the revised text as well.

- Is there some way to put Figure 5 into context, perhaps by plotting them alongside the distribution of the unmodified Mann-Kendall test or through a verbal explanation? When they are only plotted alongside one another it's hard to find the conclusion that they are 'close to one another' because there are clearly also substantial differences.

We added the unmodified Mann-Kendall test distribution on the top of the plots in Figure 6. In addition, we added to the figure the mean, variance, and skewness of each of the distributions to allow a direct comparison between the results. In the revised plot, it is more evident that all the three distributions obtained by data, ARMA method, and the modified Mann-Kendall test are sufficiently close to each other and differ from to the unmodified test. Thank you for this comment.

Specific comments:

- The title could perhaps be better phrased as 'Practical guide to using Kendall's τ ...'

Thank you for this suggestion. The title has been modified as suggested.

- Abstract, line 23: 'approaching to an impending transition' -> 'an approach of an impending transition'

We made this change to correct the typo.

- page 2 line 14: the dynamics of a system don't become slower during the approach to a tipping point, instead it's the recovery of the system from perturbations that slow down.

We modified this sentence as the reviewer suggested. The new sentence is:

“When a dynamical system approaches a tipping point, perturbations lead to long transient before the system reached its stable state, phenomenon known as slowing down.”

- p4 line 16-19: ‘Positive correlation among the observations increases the chance of obtaining a large Kendall’s τ , even in the absence of an actual trend’ is a very interesting point, and a reference to an older paper explaining this property of Kendall’s τ would be very helpful here.

We added a reference for this statement. Thank you for pointing this out.

- p4 lines 29-31: could you briefly state the dynamics that are seen in the harvesting model for this parameter – is there a single stable equilibrium, bistability etc.?

Thank you for pointing this out. The harvesting model undergoes a fold bifurcation, meaning that the system exhibits bistable dynamics for a range of parameter values in the vicinity of the tipping point. To demonstrate the system equilibria at different parameter values and the critical transition, we included more details of the harvesting model and added the bifurcation diagram of this system to the revised text. These changes are made in Section 2 and the bifurcation diagram is added to the text as Figure 1.

- p6 line 14: ‘The time series data is then down sampled to obtained another time series data with...’ -> ‘The time series data is then down sampled to obtain another time series with...’

We made this change to clarify the text. Thank you for the suggestion.

- p8 line 8: For which values of c ? Is the harvesting model stationary here or not?

The parameter value in this case is fixed at $c = 1.1$. We clarified this in the revised text.

- Figure 5: Some statistics concerning the mean, variance, and especially the skew of these distributions could strengthen your claims that they are similar.

We updated the figure and added the mean, variance, and skewness of each distribution to the plot.

- p11 lines 16-19: ‘Third, we propose that the significance of the obtained Kendall’s τ values should be studied using either the non-parametric Mann-Kendall test or parametric tests.’ should probably be reworded to emphasise that the Mann-Kendall test is a viable and possibly more practical alternative to parametric tests.

We reworded this sentence in the revised text. In particular, we noted in the revised text that:

“Third, we propose that the non-parametric Mann-Kendall test is a reliable and computationally efficient method to evaluate the significance of the estimated Kendall’s τ values revealing if a detected warning signal is a false alarm or not.”

Appendix B

To: **Andrew Dunn**
Senior Publishing Editor, Journal of the Royal Society Open Science
Re: Manuscript ID: RSOS-211346.R1

May 26, 2022

Dear Dr. Dunn,

We would like to thank you and the reviewers for the feedback. The comments of the reviewers have been very helpful in improving our paper in terms of content and readability, for which we are grateful. We have revised our paper and addressed the reviewers' comments. In particular, below are details regarding some of our specific corrections and revisions we made (shown in blue) in response to reviewers' comments.

Sincerely,
the authors

Reviewer: 1

Comments to the Author(s)

In this revised manuscript the authors propose to use the modified Mann-Kendall test for testing if a trend in an early warning signal is significant. I was rather positive, but had one important doubt, namely whether the modified test can be used in overlapping windows. Maybe I was not clear enough in my comments, but I think the authors have not dealt with this comment. They replied that indeed this is a problem for the normal Mann-Kendall test. Moreover, I suggested to compare the results with the null model method (Ebisuzaki). (In this method we generate surrogate data using a Fourier transformation with randomized phases. If the real data performs better than the surrogate data, we have a significant effect.)

The authors did reply that they did not test it but ‘believe’ that the null model will give comparable results. I found that unsatisfying and did myself a fast test to convince myself that the sliding window results of the modified Mann-Kendall test and the null model are the same (and also because I would be interested to use the modified test in future). Unfortunately I found that the modified Mann Kendall test gives much lower p values (i.e. more significant results) than the null model (especially if we use large overlapping windows). I think this must have been caused by pseudo replication due to the sliding window. Therefore I think this test cannot be used on overlapping windows. The authors could test if it works on non-overlapping windows, but probably you will then have too little data.

We agree that the stride of the moving window is an important parameter to be studied. To understand this, we analyzed four different scenarios where the total number of samples is 1200, the window size is kept at 10% (120 samples), and strides are 10, 20, 50, and 100. The results show that when there are large overlaps (10 or 20 sample stride), the normal Mann-Kendall test underestimates the variance in Kendall’s tau, while the modified Mann-Kendall test still accurately captures the distribution. However, when there are little to no overlaps between the windows (50 or 100 sample stride), both the regular and modified Mann-Kendall tests show good approximations to the true distribution. We further studied the situation where fewer samples (i.e., 120 samples) are available, and similar results were observed. Please note that the obtained results show that the modified Mann-Kendall’s test approximates the true distribution with a noticeable accuracy in all scenarios of overlapping windows that were tested. In all these examples, regardless of the percentage of overlap, the modified test did not identify statistically significant differences between the approximated Kendall’s tau and the distribution approximated by the modified test as the null hypothesis.

Nevertheless, it is worth mentioning that in the study of critical transitions, windows with large overlaps are of interest practically. The reason is that the system resilience status needs to be evaluated and updated as soon as new data is measured. Waiting for a long time to use windows with small or no overlaps to avoid the correlation might result in overlooking information regarding changes in the system resilience and impending regime shifts. In addition, as the reviewer pointed out, using non-overlapping windows requires a lot of data which might not be available in practical scenarios. This has been one of the motivations for this study, to show the issue caused by overlapping windows in practical applications, and studying the modified Mann-Kendall test as a potential solution.

We added the new results (see figure below) and details regarding overlapping windows in the revised text. We would like to thank you for this comment. It has significantly improved the paper and completeness of the analysis.

Figure 6. Comparison between the distribution of Kendall's τ approximated using data, the Mann-Kendall test, and the modified Mann-Kendall test for different window overlaps and number of observations. The control parameter is fixed at $c = 1.1$.

Reviewer: 4

Comments to the Author(s)

The paper is a reasonable explanation of potential problem with Kendall's tau for rolling window statistics. Kendall's tau has been used in the theoretical literature on early warnings of critical transitions.

However I am not aware that Kendall's tau is used by ecologists conducting whole-ecosystem field experiments. Real-world science on early warnings has used MAR models or their Bayes' analogs (e.g. Ives & Dakos, Taranu et al.), tests of heteroskedasticity (Seekell), quickest detection (Carpenter), and qualitative comparisons for example. Some papers conducted their own system-specific studies of the effects of window size. The paper provides a useful warning to ecologists as well as theoreticians but perhaps theoreticians are the primary readership.

Carpenter, S. R., Brock, W. A., Cole, J. J., & Pace, M. L. (2014). A new approach for rapid detection of nearby thresholds in ecosystem time series. *Oikos*, 123(3), 290-297. doi:10.1111/j.1600-0706.2013.00539.x

Ives, A. R., & Dakos, V. (2012). Detecting dynamical changes in nonlinear time series using locally linear state-space models. *Ecosphere*, 3(6), art58. doi:10.1890/es11-00347.1

Seekell, D., Carpenter, S., Cline, T., & Pace, M. (2012). Conditional Heteroskedasticity Forecasts Regime Shift in a Whole-Ecosystem Experiment. *Ecosystems*, 15(5), 741-747. doi:10.1007/s10021-012-9542-2

Taranu, Z. E., Carpenter, S. R., Frossard, V., Jenny, J.-P., Thomas, Z., Vermaire, J. C., & Perga, M.-E. (2018). Can we detect ecosystem critical transitions and signals of changing resilience from paleo-ecological records? *Ecosphere*, 9(10), e02438. doi:doi:10.1002/ecs2.2438

From a literature review we found that Kendall's τ is one of the most widely used metrics to detect approaching to a critical transition in the literature, particularly in climate science and ecological systems. Examples include references [7, 8, 13] of the revised text. As the reviewer points out, these studies mostly focus on surrogate ecological systems. However, researchers have also employed Kendall's tau in real-world studies of regime shifts in ecology and climate science. Below are examples of three recent studies in this field.

- Su, Haojie, et al. "Long-term empirical evidence, early warning signals and multiple drivers of regime shifts in a lake ecosystem." *Journal of Ecology* 109.9 (2021): 3182-3194.
- Syed Musa, Syed Mohamad Sadiq, et al. "An early warning system for flood detection using critical slowing down." *International journal of environmental research and public health* 17.17 (2020): 6131.
- Arkilanian, A. A., et al. "Effect of time series length and resolution on abundance-and trait-based early warning signals of population declines." *Ecology* 101.7 (2020): e03040.

We included these references in the revised manuscript. Thank you for this comment.

Reviewer: 5

Comments to the Author(s)

The manuscript by Chen et al. discusses the utility of Kendall's tau for assessing trends in indicators of critical slowing down, the importance of a significance test for the Kendall's tau estimate, and a non-parametric solution for that significance test. The topic is methodologically valuable and the authors explore the influence of various aspects of the data on the statistical estimates and tests. The exploration of moving window size and amount of data available are valuable. The authors have improved the text in response to the previous reviewers comments. There are a few additional points that can be improved upon.

I did not find the description of the moving window implementation in 2.1 complete. Was the window moved one observation at a time? Figure 1 seems to indicate movement of half the window size. This seems to be a key aspect of the resulting autocorrelation and not unambiguous. Further, do the authors mean the window sizes here as percentages of the total time series (implied by P7 reference to a window of 50% of the time series, but not actually stated that way in earlier text or figures)?

Figure 1 is only a high-level schematic of the approach, and it is not meant to reflect the window size and overlap used in the analyses. As the reviewer correctly points out, to facilitate comparison and generalization of the results, we tested different window sizes as a percentage of the total time

series and evaluated the effect of this choice on the results (Fig. 5 of the revised text). We revised the text and figure captions to clarify this point.

Regarding the sliding window, the results presented in section 2.1. were obtained by using a small stride of 100. The motivation for considering a large overlap between subsequent windows is the practical application of the method where new data should be analyzed as quickly as possible to detect risks of transitions early enough. Using a large stride places the system at risk because one has to wait longer to collect enough data, and changes in the system dynamics during this period may not be noticed. Nevertheless, we analyzed in the revised text the effect of using different stride lengths on the approximated Kendall's τ values as well as on the modified Mann-Kendall's test. Results of this analysis are demonstrated in Fig. 6 of the revised text (also presented below for your convenience). We clarified the details of the sliding window and the interpretation of the results in the revised text. Thank you for this comment.

Figure 6. Comparison between the distribution of Kendall's τ approximated using data, the Mann-Kendall test, and the modified Mann-Kendall test for different window overlaps and number of observations. The control parameter is fixed at $c = 1.1$.

Finally, what is the relationship between window size and amount of data available, in other words explain to the reader why the statement on P13 L23 holds: “We encourage the use of smaller window sizes when there is a large enough amount of data.” What is the standard practice and suggestion for window sizes and overlap?

This suggestion to use smaller window sizes is based on the results presented in Fig. 3. Results show that a large window size inflates the value of Kendall's τ calculated from the time series. This causes the distribution of Kendall's τ to become flatter, and increases the chance of false alarms. However, for smaller window sizes, the distribution becomes closer to the reference normal distribution, and that results in a more reliable judgment regarding the system status.

Identifying the optimal window size requires further study and is also system dependent. For instance, it depends on the total samples available and on how fast the system stability changes over time. However, unlike most studies of early warning signals, our analysis suggests not to use large window sizes (e.g., 50%) since that flattens the Kendall's tau distribution and increases the risk of false alarms. Similarly, we do not suggest a specific overlap, but instead emphasize that if the overlap is large, it is necessary to use a modified version of the null hypothesis, such as a modified Mann-Kendall test used in this paper, in order to capture the true distribution of the Kendall's tau with a higher accuracy (Figs. 5 and 6).

The authors find that for their model of choice a time series of length <200 “the distribution becomes almost symmetric about zero, which is associated normally with random signals”. If harvesting is measured on an annual time scale this is an unlikely amount of data to have. What does this mean for the power of Kendall's tau in many of the ecological regime shifts under consideration here? Or if this is purely a function of the other parameters in the model that authors should include some discussion on this point.

We agree with the reviewer that in many real-world ecological systems the amount of available data is limited. In fact, since Kendall's τ is a metric that is commonly used in ecological systems including in practical applications as in the references [16-18] of the revised text, the goal of this study is to analyze the effect of various parameters including the sample size on the outcome of approximated Kendall's tau values. Our results warn researchers about using this metric when not enough data are available.

It is referred to again at the end with a suggestion not to detrend, but I don't think this was explained/brought up prior to the conclusion.

Thank you for pointing this out. The analysis for detrending is not based on Kendall's tau values. For consistency, we have removed that sentence.

On page 11 the authors mention 2 parametric tests and say that they are going to compare them (L24-28), but the only test compared are ARMA with the modified Mann-Kendall.

The two parametric tests described in page 11 are a literature review of the methods that are being used. In our work, we selected one of them - the ARMA method - to compare with results obtained from non-parametric Mann-Kendall tests. We applied a minor change in the text to clarify that this section is a literature review.

Clearly the modified Mann-Kendall is more straightforward and efficient. The authors mention in the introduction that parametric tests can be more powerful. Does Figure 6 suggest similar power in this scenario or can the authors speak to that point?

Parametric tests are more powerful than standard non-parametric tests since they are based on more information from the system. While results of Figure 6 (Figure 7 in the revised text) show a slightly higher accuracy for the parametric test, this statement cannot be explained using a single example. The modification that has been applied to the standard Mann-Kendall's test has also played a significant role in improving its accuracy in the context of early warning signals.

The conclusion summarizes the findings on Kendall's tau. Nowhere in the paper are alternatives to Kendall's tau mentioned or whether the issues raised here suggest cases when Kendall's tau should not be used.

Thank you for this comment. Indeed, the focus of this work is on analyzing the parameters that affect the estimated Kendall's tau values so that researchers consider these factors when using this metric in their studies. This study provides a guideline on how to interpret the Kendall's tau values and identifies when the Kendall's tau will not provide a reliable prediction and should not be used.

We have clarified the motivation of this work in the introduction and conclusions. Thank you for this comment.

Minor points:

I found the text label "Trend: Kendall's tau" on Figure 1 to be confusing as it seemed to correspond to the addition of the single new measurement, rather than a calculation across all the indicator values.

We revised the figure and removed this text.

P2 L13 Check verb tense agreement

P3 L15 Check verb number agreement

P5 L53 "parameter" -> "parameters"

P7 L48 Check verb tense

P11 L26 Missing word?

We fixed these typos. Thank you.

We would like to thank you for your detailed comments. It has significantly improved the paper and completeness of the analyses.

Appendix C

To: **Andrew Dunn**
Senior Publishing Editor, Journal of the Royal Society Open Science
Re: Manuscript ID: RSOS-211346.R2

June 24, 2022

Dear Dr. Dunn,

We would like to thank you and the reviewers for the feedback. We applied the requested minor changes to the revised manuscript.

Sincerely,
the authors

Reviewer 5

The authors have done a nice job of responding to reviewer comments, including expanding the consideration of window sizes and overlaps. I found the new version improved and clear. I have only a couple small items I noticed:

P3 L21 Awkward sentence structure in new text

P6 L40 and Figure 2 caption: the text says "smaller window size and a large window size". The panels are in order of largest to smallest. Would be better to match these in one order or the other.

P10 L18 Is it necessary/helpful to say anything about calculating the autocorrelation? Does this reduce the degrees of freedom in the hypothesis test?

P10 L34 Is it meant to say Eq 4?

Thank you for your pointing these out. We revised our manuscript and addressed these comments.

Sincerely,
The authors